# Magnesium Alloy Matching Layer for High-Performance Transducer Applications

**DOI:** 10.3390/s18124424

**Published:** 2018-12-14

**Authors:** Yulei Wang, Jingya Tao, Feifei Guo, Shiyang Li, Xingyi Huang, Jie Dong, Wenwu Cao

**Affiliations:** 1National Engineering Research Center of Light Alloy Net Forming, School of Materials Science and Engineering, Shanghai Jiao Tong University, Shanghai 200240, China; vincent_dec@sjtu.edu.cn (Y.W.); crystal0826@sjtu.edu.cn (J.T.); 2Department of Instrument Science and Engineering, Shanghai Jiao Tong University, Shanghai 200240, China; shiyangli@126.com; 3Shanghai Key Lab of Electrical Insulation and Thermal Aging, Shanghai Jiao Tong University, Shanghai 200240, China; xyhuang@sjtu.edu.cn; 4Department of Mathematics and Materials Research Institute, The Pennsylvania State University, University Park, PA 16802, USA; wcao@hit.edu.cn

**Keywords:** matching layer, magnesium alloy, ultrasonic transducer

## Abstract

In this paper, we report the use of magnesium alloy (AZ31B) as the matching material for PZT-5H ultrasonic transducers. The AZ31B has an acoustic impedance of 10.3 MRayl, which provides a good acoustic impedance match for PZT-5H ultrasonic transducers in water medium based on the double matching layer theory. Two PZT-5H transducers with different center frequencies were designed and fabricated using the AZ31B. The respective center frequencies of the two fabricated transducers were 4.6 MHz and 9.25 MHz. The 4.6 MHz transducer exhibits a −6 dB bandwidth of 79% and two-way insertion loss of −11.11 dB. The 9.25 MHz transducer also shows good performance: −6 dB bandwidth of 71% and two-way insertion loss of −14.43 dB. The properties of the two transducers are superior to those of transducers using a composite matching layer, indicating that the magnesium alloy may be a promising alternative for high-performance transducers.

## 1. Introduction

As ultrasonic imaging is becoming increasingly important in modern medicine, higher-performance ultrasonic transducers will be required in the future [1,2]. For ultrasonic transducers, the choice of piezoelectric materials is critical [3]. Compared with piezoelectric polymers and single crystals, lead-based piezoceramics (PZT) are more suitable for transducers because of their excellent piezoelectric properties and superior temperature stability [4]. However, PZT ceramics have an acoustic impedance of 33 MRayl, which severely mismatches with that of the human body (~1.5 MRayl). The large acoustic impedance mismatch can cause about 80% ultrasound energy loss at the interface, resulting in poor sensitivity and long ringdown [5]. Based on previous works on ultrasonic transducers, one of the most effective ways to improve the properties of transducers is the development of matching layers [6,7]. As a classical model, the Krimholz–Leedom–Mattaei (KLM) equivalent circuit has been widely used in design of matching layers [8]. Therein, double quarter-wavelength matching layers are most frequently used. The optimized acoustic impedance values of the two matching layers are 8.9 MRayl and 2.3 MRayl for PZT-5H ultrasonic transducers. Traditionally, the first matching materials are composites made of solid particles/templates and polymer [9]. Tung Manh et al. [10] used a silicon–polymer composite of Spurr’s epoxy with an acoustic impedance of 9.5 MRayl as the first matching layer, and the 14.6 MHz PZT transducer showed a −6 dB bandwidth of 70% and an insertion loss of −18.4 dB. An 11.6 MHz PZT-5A transducer was reported by H.J. Fang et al. [11] using anodic aluminum oxide–epoxy 1–3 composite with an acoustic impedance of 9.5 MRayl as the first matching layer, which showed a −6 dB bandwidth of 68% and an insertion loss of −22.7 dB. Although conventional composite matching materials can improve the performance of PZT transducers to some extent, there still exist some issues—nonuniformity and high attenuation, for instance—which limit further improvement of the transducers [12]. Furthermore, it is still very difficult to achieve a composite with high acoustic impedance and low loss simultaneously [10]. Therefore, it is urgent to seek a novel matching material for use in PZT ultrasonic transducers. AZ31B has an acoustic impedance of 10.3 MRayl and a small acoustic attenuation coefficient of about 0.02 dB/mm at 7.5 MHz, which can satisfy the requirement of the first matching layer of piezoelectric transducers. We used this material in a PMN-PT (lead magnesium niobate-lead titanate) single crystal ultrasonic transducer, and the results were very promising [13]. Here, AZ31B is proposed to be applied in PZT ultrasonic transducers with the hope to improve the performance of the PZT-5H transducers. Two PZT-5H transducers with center frequencies of 5 MHz and 10 MHz were designed and fabricated, and their properties were evaluated.

## 2. Materials and Methods

### 2.1. Design of the Transducers

Figure 1 shows the design cross sections of the proposed ultrasonic transducers. As shown in Figure 1, commercial PZT-5H ceramics were selected as the active element of the transducers, and the main parameters are shown in Table 1. Taking 50 Ω electric impedance matching into account, the apertures/thicknesses of the active elements for the designed 5 MHz and 10 MHz transducers were 4.0 × 4.0 mm^2^/400 μm and 2.0 × 2.0 mm^2^/200 μm, respectively.

AZ31B and Epo-Tek 301 (Epoxy Technology Inc., Biller-ica, MA, USA) were chosen as the matching materials. The acoustic impedances of these two materials are 10.3 MRayl and 3.0 MRayl, respectively. The thicknesses of the two matching layers were one-quarter of the wavelength at the center frequency, according to the KLM model [8]. The average grain size of the AZ31B used in these transducers is about 5 μm, as shown in Figure 2, which would not affect the transmission of the sound waves at the designed frequencies. A mixture of Epo-Tek 301, tungsten powder, and glass microspheres was used as the backing, and showed an acoustic impedance of 9.12 MRayl. The mass ratio of the Epo-Tek 301, tungsten powder, and glass spheres was 4:1:0.5. The basic properties of the passive materials used in this work are summarized in Table 2. 

### 2.2. Transducer Fabrication and Characterization

In this work, the two transducers were fabricated using the same procedure as follows. Taking the 5 MHz transducer as an example, firstly, the piezoelectric element with an aperture of 4.0 × 4.0 mm^2^ and thickness of 400 μm was prepared. Both sides of the active element were sputtered with Au electrode. Secondly, the first matching layer of AZ31B was lapped down to 317 μm and polished, then the second matching layer of Epo-Tek 301 with a thickness of 145 μm was cured over the polished surface of the AZ31B. To optimize the bandwidth and sensitivity, a backing with a thickness of 5 mm was used. Thirdly, the prepared active element, two matching layers, and backing were bonded together using the Epo-Tek 301. After curing for 24 h at room temperature, two electrodes of piezoelectric element were electrically connected to the wire using conductive epoxy (E-solder 3022, Von Roll Isola Inc., New Haven, CT, USA). Lastly, the acoustic stack was fixed in a steel housing by curing Epo-Tek 301 in the void. For the 10 MHz transducer, the thicknesses of the AZ31B, Epo-Tek 301, and backing were 156 μm, 71.2 μm, and 4 mm, respectively. The sizes of the fabricated 5 MHz and 10 MHz transducers were Φ9.5 mm × 20 mm and Φ7.5 mm × 20 mm, respectively, as shown in Figure 3. 

The electrical impedance and phase measurements were taken using an impedance analyzer (4294A, Agilent, Santa Clara, CA, USA) [15] in air. The characterization of the pulse–echo responses and the insertion loss was carried out in a deionized water bath, using an X-cut quartz as target. The scheme of the experimental measurements of the pulse–echo response and insertion loss is shown in Figure 4. To measure the pulse–echo responses of the transducers, a 75 MHz pulser/receiver (5073PR, Olympus, Tokyo, Japan) with a damping impedance of 50 Ω was used to excite the transducers and receive the echoes. The echoes were digitized by an oscilloscope (TBS1102B, Tektronix, Beaverton, OR, USA), and the frequency spectra were obtained from the echo responses using Fast Fourier Transform [16]. To evaluate the two-way insertion loss, a tone burst of a 20-cycle sine wave generated by a function generator (DG4162, RIGOL, Beijing, China) with an amplitude of 1 V was used to excite the transducers, and the echoes were received by a 1 MΩ coupling oscilloscope (TBS1102B, Tektronix). The signal loss caused by the transmission into the quartz target and the attenuation in water were compensated. The value of the two-way insertion loss (IL) was calculated from [17]
(1)$$\begin{array}{c} IL=20log\frac{V_{o}}{V_{i}}+1.9+2.2\times 10^{-4}\cdot 2d\cdot f_{c}{}^{2} \end{array}$$ where *V_o_* is the output voltage amplitude of the transducer, *V_i_* is the input voltage amplitude of the excitation pulse power, *d* is the distance between the transducer and the quartz, and *f_c_* is the center frequency of the transducer. 

## 3. Results and Discussions

Figure 5 and Figure 6 display the electrical impedance and phase of the two transducers. Both transducers show two resonance peaks. The first one at lower frequency is derived from the thickness vibration mode of the piezoelectric element. Through experimental verification, the second one at higher frequency is attributed to the AZ31B matching layer. The effective electromechanical coupling factors *k_t_*_(*eff.*)_ of the two transducers are approximately equal to that of the PZT-5H ceramics, as shown in Table 3.

Figure 7 and Figure 8 show the simulated pulse–echo responses and the FFT spectra of the 5 MHz and 10 MHz transducers, respectively. According to the simulated results, the designed 5 MHz transducer shows a center frequency of 4.73 MHz with a −6 dB bandwidth of 77.38%. The other transducer with a center frequency of 9.61 MHz exhibits a −6 dB bandwidth of 77.00%. Figure 9 and Figure 10 show the measured pulse–echo responses and the FFT spectra of the 5 MHz and 10 MHz transducers. The center frequency, −6 dB bandwidth, and two-way insertion loss of the 5 MHz transducer are about 4.60 MHz, 79%, and −11.11 dB, respectively. For the 10 MHz transducer, these values are 9.25 MHz, 71%, and −14.43 dB, respectively. It can be clearly seen that the measured center frequencies of the two transducers are consistent with the simulated results. Although the measured −6 dB bandwidths of the two transducers are slightly lower than the simulated results, they are superior to those of commercial PZT transducers [18]. The most remarkable fact is that both transducers show much lower two-way insertion loss than do other reported transducers in the literature [10,11], which can be attributed to the suitable acoustic impedance and lower acoustic attenuation coefficient of the AZ31B. Overall, compared with other reported transducers using conventional composites, the fabricated PZT-5H transducers with AZ31B as the matching material show broader bandwidth and much lower insertion loss, as shown in Table 4.

## 4. Conclusions

In conclusion, we have demonstrated the feasibility of AZ31B magnesium alloy in PZT ultrasonic transducer applications. AZ31B has a suitable acoustic impedance and a much smaller acoustic attenuation coefficient compared with conventional composites. Two PZT-5H ultrasonic transducers with an AZ31B matching layer were designed and fabricated. The 4.6 MHZ transducer exhibits a −6 dB bandwidth of 79% and a two-way insertion loss of −11.11 dB. The 9.25 MHz transducer also shows excellent results with a −6 dB bandwidth of 71% and an insertion loss of −14.43 dB. Our results reported in this paper indicate that the magnesium alloy is a promising matching material for high-performance transducer applications.

## Figures and Tables

**Figure 1 sensors-18-04424-f001:**
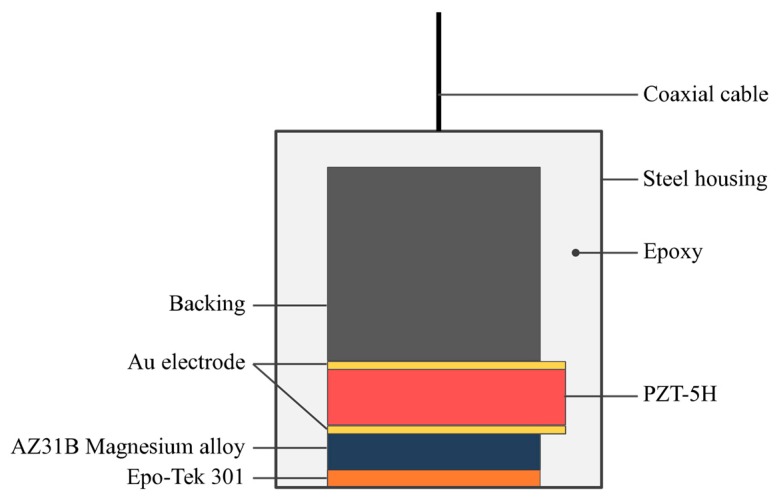
The design cross sections of the lead-based piezoceramic (PZT) transducer with an AZ31B magnesium alloy matching layer.

**Figure 2 sensors-18-04424-f002:**
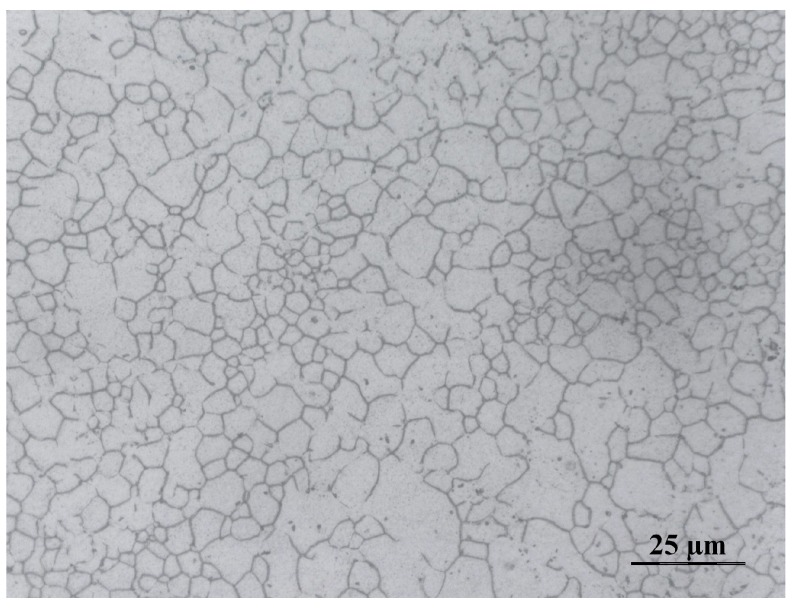
Micrograph of the AZ31B magnesium alloy.

**Figure 3 sensors-18-04424-f003:**
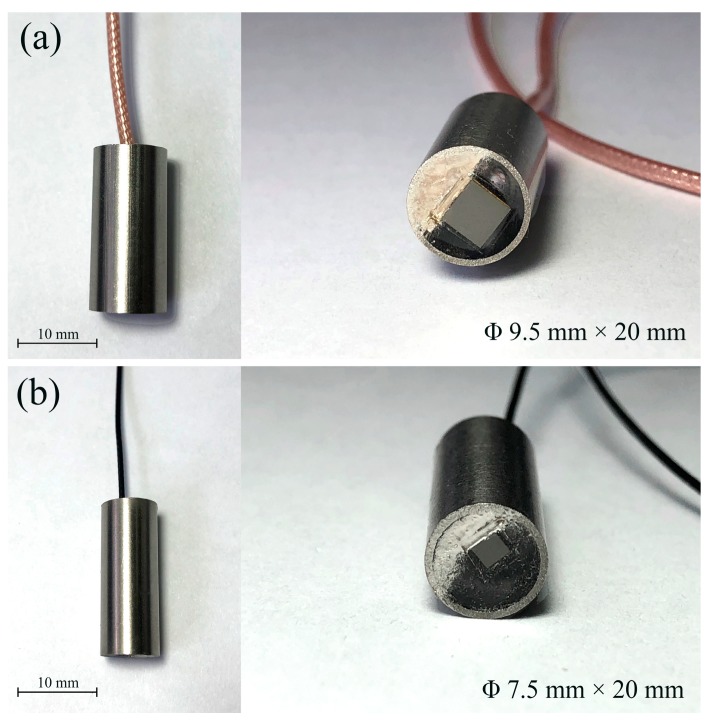
Photographs of the fabricated ultrasonic transducers: (**a**) 5 MHz, (**b**) 10 MHz.

**Figure 4 sensors-18-04424-f004:**
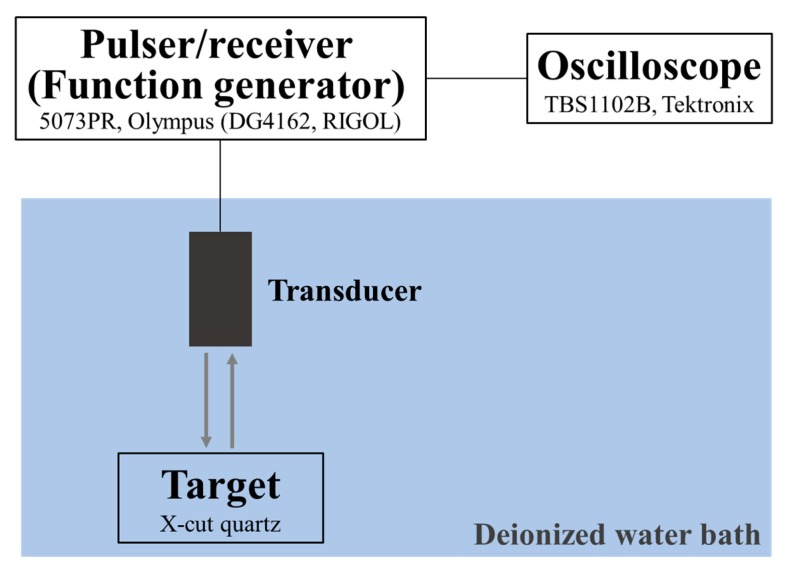
Scheme of the experimental measurements of the pulse–echo response and insertion loss.

**Figure 5 sensors-18-04424-f005:**
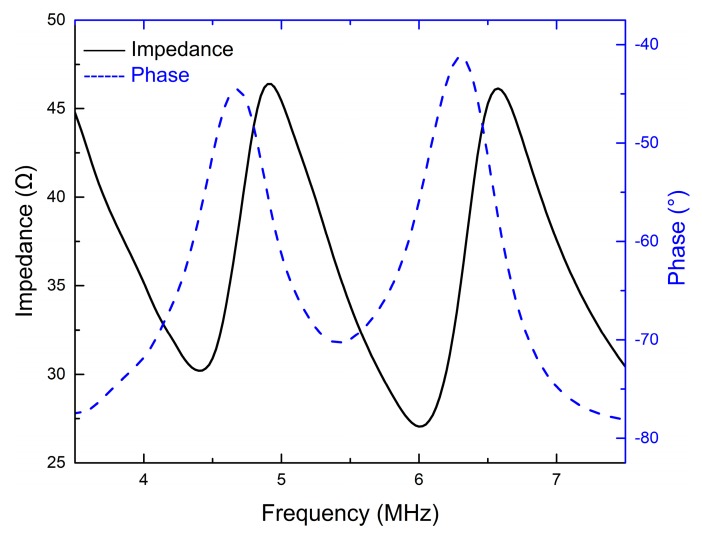
The electrical impedance and phase of the 5 MHz transducer.

**Figure 6 sensors-18-04424-f006:**
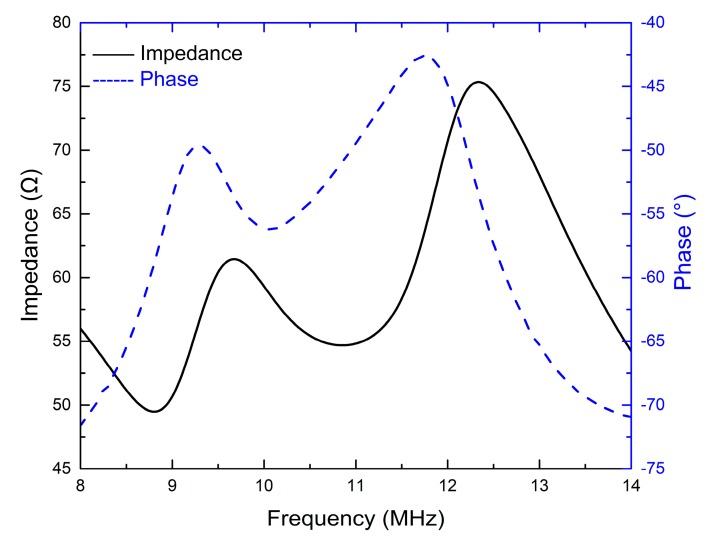
The electrical impedance and phase of the 10 MHz transducer.

**Figure 7 sensors-18-04424-f007:**
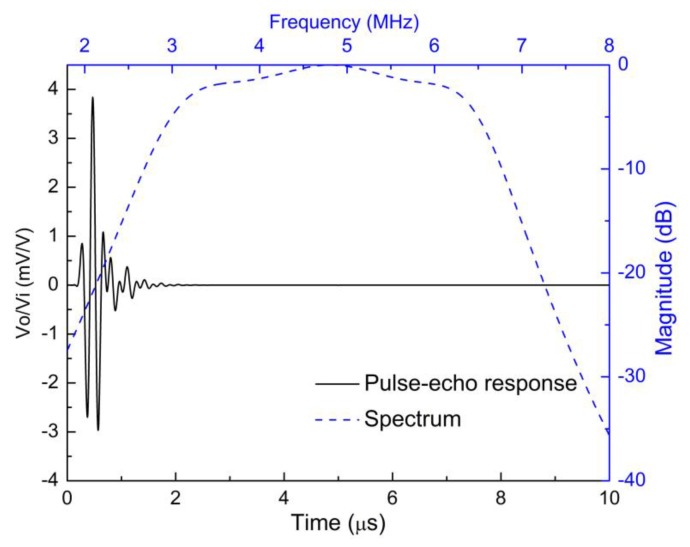
The modeled pulse–echo responses and the FFT spectrum of the 5 MHz transducer.

**Figure 8 sensors-18-04424-f008:**
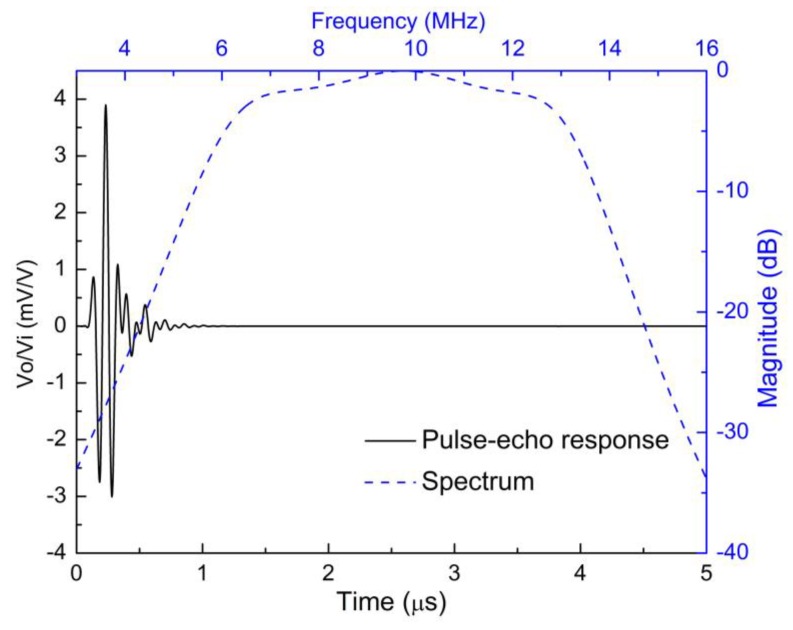
The modeled pulse–echo responses and the FFT spectrum of the 10 MHz transducer.

**Figure 9 sensors-18-04424-f009:**
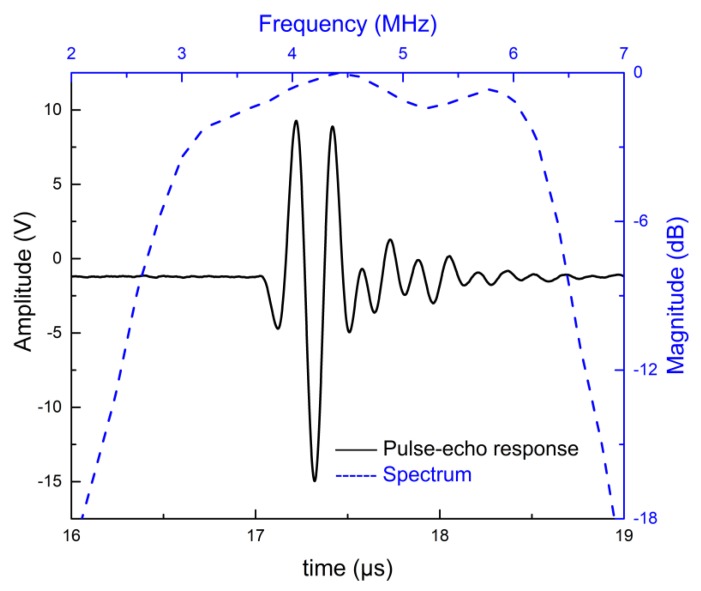
The measured pulse–echo responses and the FFT spectrum of the 5 MHz transducer.

**Figure 10 sensors-18-04424-f010:**
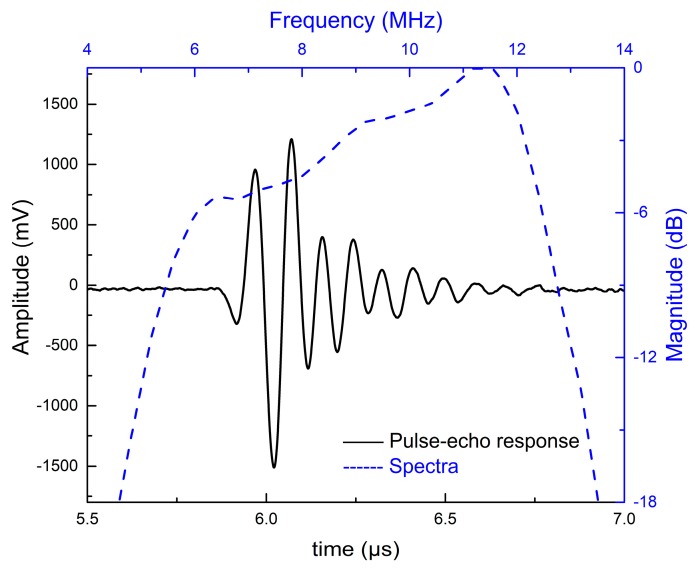
The measured pulse–echo responses and the FFT spectrum of the 10 MHz transducer.

**Table 1 sensors-18-04424-t001:** The main parameters of the PZT-5H ceramics.

	PZT-5H
Density, *ρ*	7450 kg/m^3^
Longitudinal velocity, *v*	4560 m/s
Piezoelectric constant, *d*_33_	670 pC/N
Related clamped dielectric constant, *ε^s^/ε*_0_	1802
Frequency constant (thickness mode), $N_{t}^{D}$	1989 Hz·m
Electromechanical coupling coefficient, *k_t_*	0.505
Acoustic impedance, *Z_a_*	34.2 MRayl

**Table 2 sensors-18-04424-t002:** The basic physical properties of the matching and backing layers.

Material ^1^	Use	*v* (m/s)	*ρ* (kg/m^3^)	*Z_a_* (MRayl)	Loss (dB/mm)
Tungsten/glass spheres/Epo-Tek 301 ^1^	Backing	2256	4040	9.1	N/A
Epo-Tek 301	Matching layer 2	2650	1150	3.0	9.5 (at 30 MHz) [14]
AZ31B Magnesium alloy	Matching layer 1	5800	1780	10.3	0.02 (at 7.5 MHz)
Silver–epoxy composite ^2^	Matching layer	3860	1900	7.3	13.8 (at 30 MHz)
Alumina/polymer nanocomposite films ^2^	Matching layer	3200	1630	5.1	15 (at 40 MHz)

^1^
*v* is the longitudinal sound velocity; Loss is the acoustic attenuation per unit length for longitudinal soundwaves. ^2^ The properties of a silver–epoxy composite matching layer [14] and alumina/polymer nanocomposite films [12] are cited for comparison.

**Table 3 sensors-18-04424-t003:** Measured electrical impedance results of the transducers.

Property	5 MHz Transducer	10 MHz Transducer
Resonance frequency	4.4 MHz	8.81 MHz
Anti-resonance frequency	4.87 MHz	9.67 MHz
*k_t_*_(*eff.*)_	0.43	0.41

**Table 4 sensors-18-04424-t004:** Summary of properties of the fabricated PZT-5H transducers and other reported results.

Active Element	Matching Materials	Backing Layers	*f_c_* (MHz)	*BW* (%)	*IL* (dB)
PZT-5H	AZ31B/Epo-Tek 301	Tungsten powder/glass microspheres/Epo-Tek 301	4.60	79	−11.11
PZT-5H	AZ31B/Epo-Tek 301	Tungsten powder/glass microspheres/Epo-Tek 301	9.25	71	−14.43
PZT-5A ^1^	Anodic aluminum oxide-epoxy/Epo-Tek 301	Tungsten powder/micro bubbles/Epo-Tek 301	11.6	68	−22.7
PZT-5A ^1^	silicon–polymer 1–3 composite/Epo-Tek 301	Tungsten powder/micro bubbles/Epo-Tek 301	15	50	-
PZT ^2^	2-2 silicon–polymer composite/Spurr’s epoxy	Air	14.6	70.2	−18.4

^1^ Cited from Fang H.J. et al.’s results [11]. ^2^ Cited from Manh T. et al.’s results [10].

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
