# Peer review of "Magnesium Alloy Matching Layer for High-Performance Transducer Applications"

_sensors, 2018, doi:10.3390/s18124424_

Round 1

Reviewer 1 Report

Comments to the Author,

 1. Comment #1 : As the authors published the previous results of using 3 magnesium alloy matching layers for PMN-PT, it would be very kind to provide some similar data for applying the other magnesium alloy matching layers such as GW83 and ZK60 to the PZT-5H.

 2. Comment #2 : In the abstract, it would be very kind to eliminate qualitative words such as “quite low” and “much superior”. Instead of those words, please provide a quantitative comparison. Also, please replace all the qualitative words with the quantitative words in the introduction as well.

 3. Comment #3 : In the introduction, the authors briefly mentioned the non-uniformity and high attenuation problems of the conventional composite matching materials. Could you please explain more about the non-uniformity issues?

 4. Comment #4 : In figure 6, 7, 8, and 9, it would be very kind to fix the scale of the y-axis constantly. Mainly, I would prefer seeing the figure 8 and 9 with the same scale as the right y-axis of the figure 6 or 7. It would be clear to calculate -6 dB bandwidth with the same y-axis scale.

Author Response

1. Comment #1 : As the authors published the previous results of using 3 magnesium alloy matching layers for PMN-PT, it would be very kind to provide some similar data for applying the other magnesium alloy matching layers such as GW83 and ZK60 to the PZT-5H.

Response: Thank you for your suggestion. We focused on the advantage of the AZ31B magnesium alloy as the matching layer of PZT transducers in this work. We will try to fabricate PZT-5H transducers with GW83, ZK60 as the matching layer in further researches.

2. Comment #2 : In the abstract, it would be very kind to eliminate qualitative words such as “quite low” and “much superior”. Instead of those words, please provide a quantitative comparison. Also, please replace all the qualitative words with the quantitative words in the introduction as well.

Response: Quantitative comparisons had been shown in Table 3. The qualitative words have been all deleted in the revised manuscript.

3. Comment #3 : In the introduction, the authors briefly mentioned the non-uniformity and high attenuation problems of the conventional composite matching materials. Could you please explain more about the non-uniformity issues?

Response: The non-uniformity refers to the uniform distribution of metal powders in the polymer matrix, which results in differences in density and velocity throughout the matching layer.

4. Comment #4 : In figure 6, 7, 8, and 9, it would be very kind to fix the scale of the y-axis constantly. Mainly, I would prefer seeing the figure 8 and 9 with the same scale as the right y-axis of the figure 6 or 7. It would be clear to calculate -6 dB bandwidth with the same y-axis scale.

Response: Thank you for your advice. We have fixed the scale of the y-axis constantly in Fig. 7,8,9 and 10 in the revised manuscript.

Reviewer 2 Report

The literature widely presents the results of theoretical and experimental studies on the issue discussed in the article. Using known methods of calculation of piezoelectric transducers, the authors selected suitable materials for the creation of a broadband transducer. Thus, the authors were able to add an element of novelty to the issue under discussion by experimentally confirming the applicability of the selected materials. This is the value of the presented article. The work may be of interest to developers of broadband transducers.

According to the reviewer, the article should be revised. The following additions should be made to the article:

1. Simultaneous use of the damper ("Backing" in Fig. 1) and the matching layers do not allow to fully judge the effect of the use of matching layers. It is necessary to show separately the role of the damper and the role of matching layers. For this purpose it is necessary to provide data on bandwidth of the transducer with the damper (without matching layers). The damper must be the same as that used in the described experiment. For comparison, then you need to show the same data for the transducer with the same damper and matching layers. If it is technically possible, the case in which the transducer does not have a damper, but has only matching layers, would be of interest.You can also specify pulse durations for these cases.

2. It is necessary to specify the mass fractions of the damper components or at least its effective specific acoustic impedance. Otherwise, it is difficult to judge the role of matching layers to extend the bandwidth of the transducer.

3. It is advisable to give a scheme of the experimental measurements (and possibly bringing photos).

4. It is necessary to show the overall dimensions of the transducers (in Fig. 3).

5. It is known that the sound attenuation coefficient in materials depends, in particular, on the frequency and grain of the material. Table 2 lists Loss (dB/mm) for AZ31B Magnesium alloy. What frequency? What is the average grain size of AZ31B Magnesium alloy? These data must be specified in the work.

Author Response

1. Simultaneous use of the damper ("Backing" in Fig. 1) and the matching layers do not allow to fully judge the effect of the use of matching layers. It is necessary to show separately the role of the damper and the role of matching layers. For this purpose, it is necessary to provide data on bandwidth of the transducer with the damper (without matching layers). The damper must be the same as that used in the described experiment. For comparison, then you need to show the same data for the transducer with the same damper and matching layers. If it is technically possible, the case in which the transducer does not have a damper, but has only matching layers, would be of interest. You can also specify pulse durations for these cases.

Answer: Some other researchers has compared the performance of transducers with or without matching layers. Brown J. , for example, reported a transducer with two matching layers with a bandwidth of 59% and insertion loss of -18.2 dB, while the transducer without matching layers only showed a bandwidth of 25% and insertion loss of -34.1 dB .

Reference: Brown, J.; Sharma, S.; Leadbetter, J.; Cochran, S.; Adamson, R. Vacuum deposition of mass-spring matching layers for high-frequency ultrasound transducers. In 2014 IEEE International Ultrasonics Symposium; 2014; pp. 101–104.

In our previous article, we fabricated PMN-PT single crystal transducers with different matching layers using the same backing. The bandwidth and the insertion loss of the transducer with AZ31B magnesium alloy matching layer were 67% and -11.4 dB, much superior to the transducer with the conventional matching layer, with a bandwidth of 40% and insertion loss of -13.5 dB.

Reference: Guo, F.; Wang, Y.; Huang, Z.; Qiu, W.; Zhang, Z.; Wang, Z.; Dong, J.; Yang, B.; Cao, W. Magnesium Alloy Matching Layer for PMN-PT Single Crystal Transducer Applications. IEEE Trans. Ultrason. Ferroelectr. Freq. Control 2018, 65, 1865–1872, doi:10.1109/TUFFC.2018.2861394.

Based on the above, we fabricated PZT-5H transducer with AZ31B magnesium alloy as the matching layer in this work, and good performance of the transducer could be expected.

2. It is necessary to specify the mass fractions of the damper components or at least its effective specific acoustic impedance. Otherwise, it is difficult to judge the role of matching layers to extend the bandwidth of the transducer.

Answer: The acoustic impedance of the backing is 9.1 MRayl, as listed in Table 2. The mass ratio of the Epo-Tek 301, tungsten powder and glass spheres are 4:1:0.5. And the mass fractions have been added in the revised version of the manuscript, see Section 2.1, paragraph 2, line 8.

3. It is advisable to give a scheme of the experimental measurements (and possibly bringing photos).

Answer: The scheme of the experimental measurements has been added to the revised version as Fig. 4 in the revised version of the manuscript.

4. It is necessary to show the overall dimensions of the transducers (in Fig. 3).

Answer: Thank you for reminding. The size of the fabricated transducer had been added to the revised manuscript, see Section 2.2, paragraph 1, line 13.

5. It is known that the sound attenuation coefficient in materials depends, in particular, on the frequency and grain of the material. Table 2 lists Loss (dB/mm) for AZ31B Magnesium alloy. What frequency? What is the average grain size of AZ31B Magnesium alloy? These data must be specified in the work.

Answer: The frequency of the loss measurement has been specified in the text (see Section 1, paragraph 1, line 27) and Table 2.

The average grain size of AZ31B Magnesium alloy had been mentioned in Section 2.1, paragraph 2, line 4 when explaining the matching layer materials.